# Validation of Loop-Mediated Isothermal Amplification (LAMP) Field Tool for Rapid and Sensitive Diagnosis of Contagious Agalactia in Small Ruminants

**DOI:** 10.3390/ani10030509

**Published:** 2020-03-19

**Authors:** Serena Tumino, Marco Tolone, Alessio Parco, Roberto Puleio, Giuseppe Arcoleo, Claudia Manno, Robin A.J. Nicholas, Guido Ruggero Loria

**Affiliations:** 1Dipartiment of Agricultural, Food and Environmental Science, University of Catania, 95123 Catania, Italy; 2Istituto Zooprofilattico Sperimentale della Sicilia “A. Mirri”, 90129 Palermo, Italy; marco.tolone@unipa.it (M.T.); alessio.parco@gmail.com (A.P.); roberto.puleio@izssicilia.it (R.P.); guidoruggero.loria@izssicilia.it (G.R.L.); 3Department of Agricultural, Food and Forestry Science, University of Palermo, 90128 Palermo, Italy; 4Enbiotech SRL, 90129 Palermo, Italy; g.arcoleo@enbiotech.eu; 5AVANTECH GROUP SRL, 84012 Angri, Italy; manno@avantech.it; 6The Oaks, Nutshell Lane, Farnham, Surrey GU9 0HG, UK; robin.a.j.nicholas@gmail.com

**Keywords:** *Mycoplasma agalactiae*, field diagnostic test, *p40* gene, small ruminants, LAMP

## Abstract

**Simple Summary:**

Contagious agalactia (CA) is an infectious disease of small ruminants endemic in the Mediterranean countries, causing significant socioeconomic impacts predominantly on small-scale farmers who still subsist on marginal lands. *Mycoplasma agalactiae* is historically considered the principal etiological agent of CA, especially in sheep. Clinical signs are characterised by mastitis, arthritis, keratoconjunctivitis and occasionally, abortion. Rapid, accurate and cost-effective field tests are urgently needed for effective control of *M. agalactiae* mastitis. Our study illustrated the validation of a Loop-Mediated Isothermal Amplification (LAMP) test for the detection of *M. agalactiae* in dairy sheep in order to confirm its application as a diagnostic tool in the field level.

**Abstract:**

Contagious agalactia (CA), an infectious disease of small ruminants, caused by *Mycoplasma agalactiae*, is responsible for severe losses to dairy sheep production with substantial socioeconomic impacts on small-scale farmers. The diagnosis of CA is still problematic, time-consuming and requires well-equipped labs for confirmation of outbreaks. Therefore, rapid, accurate and cost-effective diagnostic tests are urgently needed. This work aims to validate a novel Loop-Mediated Isothermal Amplification (LAMP) test, based on the *p40* target gene, for the detection of *M. agalactiae* in dairy sheep in order to confirm its potential practical use as a rapid and cheap field test. The LAMP system proposed in this study consists of a portable device composed of real-time fluorometer with the automatic interpretation of results displayed in a tablet. A total of 110 milk samples (90 positives and 20 negatives) were analysed to optimise the analysis procedure and to investigate the efficacy and robustness of the LAMP method. All samples were analysed using LAMP and conventional real-time PCR to compare the diagnostic sensitivity of the methods. The sensitivity of the LAMP was 10-fold higher than that of real-time PCR, with a detection limit up to 10^3^ CFU/ml. The LAMP assay was able to detect *M. agalactiae* in 81 of 90 (90%, 95%CI 0.84–0.96) positive milk samples compared to 69 (77%, 95%CI 0.59–0.95) positive samples detected by real-time PCR; no positive signal occurred for any of the negative milk samples in either test. Therefore, the LAMP assay was found to be more sensitive than real-time PCR, low-cost, easy to perform, fast and not affected by contamination, indicating its potential as an effective diagnostic tool in the field level for the diagnosis of CA.

## 1. Introduction

Contagious agalactia (CA) is an infectious disease of small ruminants endemic in the Mediterranean countries, and is commonly caused by *Mycoplasma agalactiae*. It is responsible for severe losses to dairy sheep production with substantial socioeconomic impacts on small-scale farmers, often subsisting on marginal land.

Clinical signs are characterized by mastitis, arthritis, keratoconjunctivitis and occasionally, abortion. The course of the disease starts as mild monolateral interstitial mastitis, with warm, swollen, painful parenchyma followed by sclerosis of the udder, alteration in the quality of milk with a drastic decline in dairy production [1,2,3].

Today in the Middle East and countries in the Mediterranean basin, the management of CA is one of the highest priorities for sheep and goat farming due to the severe losses in milk production, increased lamb mortality, cost of veterinary assistance and the difficulty of eradicating the infection once established in a herd. Considering the widespread distribution and economic impact, CA has been included as a notifiable disease listed by the OIE (World Organization for Animal Health) [1].

Early diagnosis is essential for the rapid and effective management of the disease in order to avoid the spread of the infection to the whole flock; therefore, rapid, precise, and low-cost methods for pathogen detection are urgently needed. Traditionally, according to OIE guidelines [1], the current diagnosis of CA is based on the isolation of causative mycoplasmas from affected animals, which are further identified by biochemical, serological or molecular tests such as PCR-based methods. The isolation of the mycoplasma in selective enrichment media represents a laborious and time-consuming process, as mycoplasmas grow very slowly. Furthermore, serological tests such as the enzyme immunoassay or complement fixation test may be ineffective in the first stage of the disease, which may result in false-negative cases because the antibodies are detectable only after 10–15 days from infection [4]. During the past decades, a number of polymerase chain reaction (PCR) and real-time quantitative PCR-based assays have been widely applied for rapid detection of *M. agalactiae* [5,6,7]. Nevertheless, disadvantages for PCR and real-time PCR such as the presence of inhibitors normally in milk and the requirement for trained staff and well-equipped laboratories limit their utilization as rapid tests and in-field practice. More recently, new molecular diagnostic tools have been developed to provide higher sensitivity, specificity and to reduce time and costs. Loop-Mediated Isothermal Amplification (LAMP) is an innovative and economic gene amplification tool based on its ability to amplify a target gene with high efficiency under isothermal conditions, unlike PCR, in the range of 60 to 65 °C [8]. In comparison to the conventional PCR and real-time PCR, the LAMP technique is less sensitive to inhibitors present in biological samples. It does not require temperature cycling and can be performed using a simple heating device such as water bath; therefore, it can potentially be used for tests in the field [9]. LAMP is more specific and faster than PCR and real-time PCR because it employs four oligonucleotide primers, namely FIP (forward inner primer), BIP (backward inner primer), F3 (forward primer) and B3 (backward primer) to recognize six different regions of the target gene. Two extra primers, LF (loop forward) and LB (loop backward), are also incorporated in order to accelerate the amplification of reaction as well as enhance the specificity [10]. LAMP has been shown to be a sensitive and specific method for the detection of veterinary pathogens [11,12,13,14]. The detection of *M. agalactiae* by LAMP was first reported by Rekha et al. [15]. However, the method for the detection of *M. agalactiae* in milk, the most common sample received by laboratories, has not yet been validated on field samples.

The aim of this work is to validate a LAMP test, previously developed by Loria et al. [16] for the detection of *M. agalactiae* in sheep milk samples in order to confirm both its effectiveness and robustness as a diagnostic tool and its potential practical use as a rapid and cheap field test. 

## 2. Materials and Methods 

### 2.1. Samples Preparation

The NCTC reference strain of *M. agalactiae* (NCTC 10123) and three wild strains of *M. agalactiae* (Sc 123/4; Pa 116/20; Pa 49/19), previously isolated and identified by the OIE Reference Laboratory for CA at the Istituto Zooprofilattico Sperimentale della Sicilia, were used in this study for the artificial contamination of milk. The wild strains of *M. agalactiae* have been isolated from milk of sheep and goats affected by interstitial mastitis. In detail, the etiological agent has been identified by standard laboratory approaches according to the World Organization for Animal Health—microbiological isolation [1] and denaturing gradient gel electrophoresis (DGGE) analysis [5] of DNA extracted from 1 mL of milk.

*M. agalactiae* strains were grown in modified Hayflick’s medium and after incubation at 37 °C in 5% CO_2_ for 72 hrs, the broths were then stored at −80 °C. *M. agalactiae* cultures at exponential growth were serially diluted and colony-forming units (CFU) were determined using standard procedures [17].

To assess the limit of detection (LOD) of the assay, pasteurized milk, previously checked to be DNA mycoplasma-free by DGGE analysis, was contaminated with *M. agalactiae* and serial dilutions were made in the range of 10^7^ to 10 CFU/ml, according to Oravcovà et al. [6]. Overall, 60 previously pasteurized milk samples were spiked with a serial dilution of different wild strains of *M. agalactiae* to compare sensitivity between real-time PCR and LAMP. In addition, 30 different positive milk samples collected from Sicilian outbreaks and 20 negative samples were included in the test. The use of mycoplasma isolates from sheep and goats milk samples provided the highest level of clonal diversity for test specificity validation and the study results are valid for the source population and target population. Therefore, a total of 110 samples of milk (90 positives and 20 negatives) were used to investigate the efficacy of the new method. The genomic DNA used for the real-time PCR was extracted from 1 mL of positive and negative milk samples, using the InstaGene Matrix (Bio-Rad laboratories, Hercules, CA, USA) according to the manufacturer’s instructions and our standard laboratory approach. For the Lamp assay, the DNA extraction was performed from 100 µl of the milk sample, using a single step and an incubation at room temperature, using the reagents supplied for the kit. The conduct and reporting of each test were done blind to the other test results.

### 2.2. TaqMan Real-Time-PCR

We have compared the LAMP assay to a well-established real-time-PCR method. A TaqMan real-time PCR assay was run on a Bio-rad CFX96 real-time PCR detection system and it was performed to amplify the *p40* gene, using MAP40127F and MAP40235R primers together with a 6-carboxyfluorescein [FAM]-labelled MAP40160P probe, according to Oravcovà et al. [6].

An exogenous internal positive control (IPC) was incorporated into the PCR reaction mixture in order to assess the absence of the PCR inhibitors, according to the manufacturer’s instructions (TaqMan® Exogenous Internal Positive Control Reagents—VICTMProbe—Applied Biosystem, Foster city, CA, USA).

### 2.3. Lamp Assay

The LAMP assay was carried out using the ICGENE device (Enbiotech Group s.r.l., Palermo, Italy). It is composed of a portable instrument and a kit developed for specific detection of *M. agalactiae*. The system is composed of a real-time fluorometer with the automatic interpretation of results by the direct visualization of the sigmoid curve in a tablet, using a specific application (Enbiotech Group s.r.l., Palermo, Italy).

Six primers targeting the p40 gene [15] were used (Table 1). Three μL of the extracted DNA samples was used to obtain the specific amplification of the target in a final volume of 55 μL, including 22 μL of LAMP mix (Enbiotech, Palermo, Italy) containing freeze-dried primer and a master mix with reagents useful to carry out the test and 30 μL of mineral oil. All reagent compositions are protected by trade secret. The optimal conditions for amplification were obtained at a temperature of 64 °C for 60 min. Positive and negative DNA control included in the kit were also used. The positivity was assessed graphically by the direct visualization of the sigmoid curve on the device display.

### 2.4. Statistical Analysis

Statistical analyses were performed using the caret package in R [18]. Sensitivity (Se) was calculated as the proportion of samples identified as positive by the assay. Specificity (Sp) was calculated as the proportion of negative test results obtained among healthy controls. We also carried out a comparison of accuracy in terms of prevalence (P), positive predictive value (PPV) and negative predictive value (NPV) for real-time-PCR and LAMP methods. The study was performed in compliance with the Standards for Reporting of Diagnostic accuracy (STARD) statement [19]. 

## 3. Results and Discussion

For the evaluation of performance characteristics of LAMP assay, results were compared to those of the real-time PCR. As expected, the results showed that no positive signal occurred for any of the negative milk samples for both tests, confirming the specificity of the primer set for *M. agalactiae* reported in the literature [15,16].

The LAMP assay was able to detect *M. agalactiae* in 81 of 90 positive milk samples vs. 69 positive samples detected by real-time PCR; therefore, LAMP technology was found to be more sensitive than real-time PCR with a sensitivity of 90% (95% CI 0.84–0.96) and 77% (95% CI 0.59–0.95), respectively (Table 2). Although the sample size for test validation was not calculated a priori, considering 110 samples it was possible to obtain a test power of 90% (sensitivity).

The sensitivity of LAMP assays and real-time PCR was determined in terms of CFU by making 10-fold serial dilutions of milk contaminated with *M. agalactiae*. Real-time PCR was able to detect up to the level of 10^4^ CFU/ml, while LAMP could detect up to 10^3^ CFU/ml, indicating that LAMP was 10-fold more sensitive than real-time PCR. The processing time of the LAMP assay was within 45 min from the beginning of the amplification, rather more than the 55 min required for the real-time PCR (Figure 1). The NPV of the LAMP test was higher than the real-time PCR method (0.69% vs. 0.49%). 

Current diagnostic methods, using sheep and goats milk samples, include a variety of *M. agalactiae*-specific PCR-based assays [20]. In literature, these methods all extensively reported to have a high sensitivity and accuracy with a detection limit between 10 to 350 CFU/ml [6,7,21,22]; a quite high level in comparison to our detection limits for both the methods (real-time PCR and LAMP). Furthermore, our LAMP LOD was not as good as those reported by other LAMP assays for the detection of other veterinary mycoplasma spp. [14,15,23,24]. However, it is known that performances of molecular diagnostic methods highly depend on the DNA extraction methods used and their efficiency to remove the natural inhibitors, which are present in the milk samples [25,26]. 

Furthermore, the PCR-based assays require specific and expensive reagents, instruments and special precautions; thus, they are not suitable for application use in field, at great distances from diagnostic laboratories. On the other hand, The LAMP system with the mini portable instrument is inexpensive, requires little equipment and technical support, and is not space-consuming. In terms of turnaround time speed, the total time to detection, including the DNA extraction step, was only 1 hr and 20 min using the ICGENE device, in comparison with 2 hrs of real-time PCR procedure.

Moreover, the real-time PCR price per test (in euro, excluding Value-Added Tax) was estimated to be 24 € compared to 15 € (excluding Value-Added Tax) of the LAMP test price. 

Recently, a variety of isothermal amplification methods have been developed in the molecular diagnosis of a range of diseases and play a significant role in monitoring and controlling the spread of local epidemics in several countries [27]. Each of these existing techniques has advantages and disadvantages. However, it is known that the indirect detection methods like turbidity or also the colour change resulting from the use of hydroxynaphthol blue [28,29] could be challenging to see by the naked eye. Other methods require opening the tube after amplification, such us running product on a gel and the addition of fluorescent dye (SYBR Green), causing carry-over contamination and detection of false-positive results [30,31,32]. In our experience, the best solution to avoid the risk for amplicon contamination is to use fluorescent dyes, detectable by amplification curves in real-time.

In our study, the ICGENE device has the advantage of being a portable device detecting the fluorescence emitted from the sample in real-time with the automatic interpretation of final results, observing the amplification curves on the tablet screen. Therefore, it is suitable for use directly in the field, and it does not require any process after thermal incubation, reducing the risk of environmental cross-contamination. 

## 4. Conclusions

The results obtained confirmed that the LAMP assay is faster and has more sensitivity (90% vs. 77%) than the real-time PCR method. In conclusion, the LAMP portable device could be a potential field test, because it doesn’t need the use of expensive laboratory equipment, does not require qualified staff and it is not affected by contamination.

Undoubtedly, speed, easiness and cost-effectiveness of the LAMP assay make it a promising and effective diagnostic tool in the field level for controlling *Mycoplasma agalactiae* infection.

## Figures and Tables

**Figure 1 animals-10-00509-f001:**
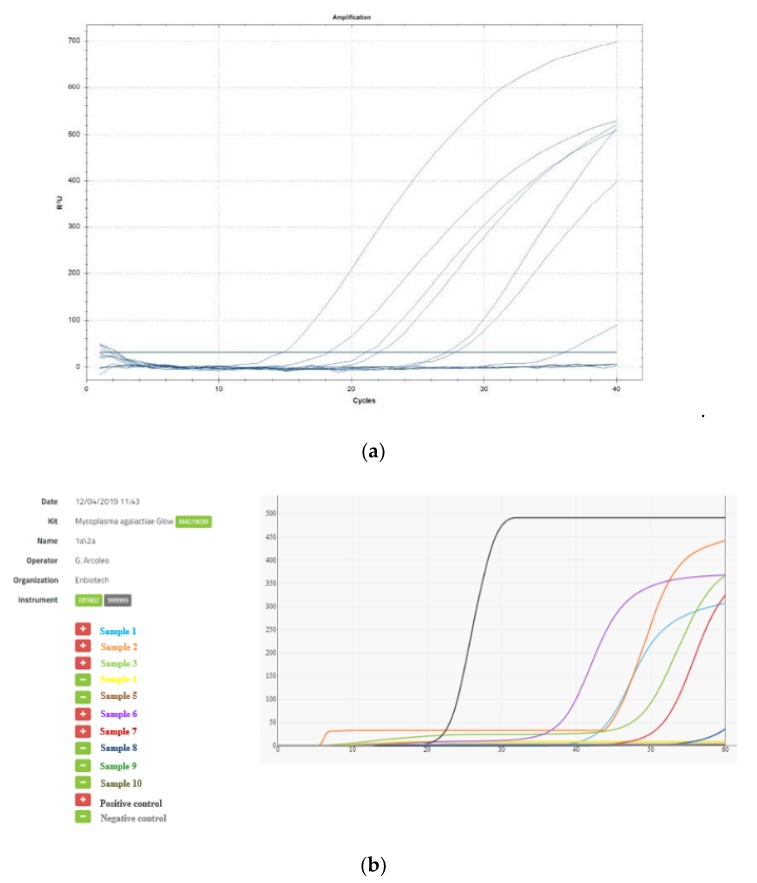
(**a**) Amplification curves obtained from real-time PCR and (**b**) LAMP assay.

**Table 1 animals-10-00509-t001:** Primer sequences for Loop-Mediated Isothermal Amplification (LAMP) for detection of *Mycoplasma agalactiae*.

Primers	Length (bp)	Primer Sequence (5′-3′)
F3	21	GGTTTATTAACTGCGTCATCA
B3	19	CAACAGTTGCATTCGTCTT
FIP	46	ACCTTATCACCATTATCTTGTGGATCAGTGCCTTTATTAGCTCGTA
BIP	46	AGCATTAGGTGAAGTTGTCAAAAATATTGAGCTTGCTTCAGGAATT
LF	24	GTGAATTTTCGTTCTTATCATCAC
LB	26	ACAAATCTAGGTGAAATAGTATTACC

**Table 2 animals-10-00509-t002:** Comparison of real-time PCR and LAMP results.

Test	TP	FP	FN	TN	P (%)	Se (95% CI)	Sp (95% CI)	PPV	NPV (95% CI)
Real-time PCR	69	0	21	20	0.82	0.77 (0.59–0.95)	1	1	0.49 (0.44–0.54)
LAMP	81	0	9	20	0.82	0.90 (0.84–0.96)	1	1	0.69 (0.65–0.73)

TP: true positive; FP: false positive; FN: false negative; TN: true negative; P: Prevalence; Se: sensitivity; Sp: specificity; PPV: positive predictive value; NPV: negative predictive value.

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
