# Peer review of "Validation of Loop-Mediated Isothermal Amplification (LAMP) Field Tool for Rapid and Sensitive Diagnosis of Contagious Agalactia in Small Ruminants"

_animals, 2020, doi:10.3390/ani10030509_

Round 1

Reviewer 1 Report

I considered the authors have clearly addressed the comments submitted in the previous review. However, I still have some minor issues regarding the manuscript:

In lines 197-199 the authors comment as a disadvantage the “indirect detection”, and include a comment about intercalating dyes such as SYBR Green or EvaGreen, however the disadvantage is not clearly explained and discussed in the manuscript. How was the amplified product detected in this manuscript? How does the test used differ from those where intercalating dyes were used?

The authors cite QPCR as an abbreviation for real time quantitative PCR on line 72, however, throughout the text use they the term real time PCR. Please use the same terminology in whole document for better clarity.

In the lines 188 to 193, the authors discuss the benefits of qPCR and LAMP, focusing on the ability of the LAMP test described, where the amplified products were detected by fluorescence, to be used as a field test. However, a discussion should be critical, and therefore, other strategies for field-based detection such as LFD, a technique previously used for the detection of LAMP tests, should be included.

The English language should be checked thoroughly in the new text added to the document (mainly lines 170-173 and lines 179-193), as there are some grammar incongruences that should be addressed.

Lines 188-193. Are LAMP reagents cheaper than those used for qPCR? Is that a real difference between qPCR and LAMP? Please clarify.

The authors do again focus on the device, however there any other commercial devices based of fluorescent detection of LAMP products. As an example, the authors cite qPCR but are not citing Biorad CFX96 throughout the text, which makes sense, so why is the ICGENE device cited so often and so specifically? As it is just a portable fluorimeter for the detection of amplified LAMP products, as the Biorad CFX96 is just a qPCR thermocycler.

Lines 191-193. Please check this sentence, as it seems like there is a missing a word.

Lines 281-285. It looks like the reference starting 31.Tatay-Dualde, J….. is misplaced, please correct.

Reviewer 2 Report

Dear authors,

The work you presented is well made with important diagnostic applications on the field. It is my pleasure to communicate with you that I accept the proposal of your manuscript in the present form.

Author Response

We're thankful for your valuable time and useful contribution.

Kind regards 

Serena Tumino

This manuscript is a resubmission of an earlier submission. The following is a list of the peer review reports and author responses from that submission.

Round 1

Reviewer 1 Report

The research presented in this manuscript aims to fill one of the most relevant knowledge gaps in contagious agalactia research. The research, development and validation of reliable point of care tests with high sensitivity and specificity is highly relevant for infectious diseases of animals, especially those notifiable, as the case of contagious agalactia.

The paper describes a solid methodology of research, however there are some aspects of the manuscripts that should be improved.

The results and discussion section is too short. There is no comparison to previous molecular tests for the detection of M. agalactiae and the description of the results is extremely brief. Therefore, this section need improving prior publication of the manuscript.

The novelty of this manuscript resides on the validation of milk samples for M. agalactiae DNA detection. However, limited information regarding the methodology for DNA extraction. This information is essential to discriminate between a potential field test, as the author claim, or just another lab-based DNA detection test slightly faster that qPCR.

Also, I have some concerns regarding the limit of detection evaluation. Oravcova et al. qPCR describes limits of detection in raw sheep going down to 10 genome equivalents. Some authors correlates mycoplasma spp. CFU values from 10 to 1000 genome equivalents. Therefore, a detection of 103 CFU could represent 10000 genome equivalents, a quite high level of detection for a molecular, even in milk. This level of detection in even higher for qPCR (hence the claim that the LAMP test described is more sensitive that qPCR). However, based of Oravcova et al. description, their PCR is able to achieve the lowest technical detection limit for a qPCR. The authors should discuss extensively their findings to account for this discrepancy.

Also, I think the discussion should focus on the nature of the detection of the device, not on the commercial name, as based on the data presented in the manuscript, that fact is not relevant. If the device detects the fluorescence emitted by the amplified product (based on intercalating dyes I presume), there are other commercial devices providing the same function in the market. I would highly recommend rewording this part of the manuscript as it could be seen as a commercial. Also, as at least one of the authors belong to the commercial company commercialising the detection device (Enbiotech SRL), this fact should be described in the conflict of interest section.

In addition, as a final suggestion to the authors, I think the manuscript would have greatly benefited from the evaluation of both goat and sheep milk, although it is true that due to the composition of the milk of sheep, it is more problematic for molecular diagnostic tests.

Other comments:

Line 20-21. The sentence starting “It is one…” sound redundant, and in addition, it focusses its relevance on dairy sheep production, however CA is highly relevant in goat herds. The sentence needs rephrasing.

Line 90. The term novel should be removed from this line, asit could be misleading. This manuscript validates a previously described method, therefore not novel.

On line 105 you describe that previously pasteurized milk was used for spiking with the pathogen. As not all thermal treatments eliminate the DNA present in the samples, did you previously check that the milk sample was M. agalactiae DNA free? That should be performed as contamination will directly affect some of the results defined in this manuscript.

Line 107. Which was the inclusion criteria for a positive sample? Were the inclusion criteria based on clinical or laboratory diagnosis? Were all the clinical samples obtained from animal with the physiological conditions? I think the origin of the clinical samples (positive and negative) should be better described.

Line 113 and 114. I think it is unnecessary to cite commercial names on these two lanes, mainly when the LAMP M. agalactiae assay is not displayed as commercially available on ICGene webpage, and you cite Bio-Rad, but there is no reference about the thermocycler used as part of your research. I would rephrase these sentence.

Lane 123. The LAMP reaction should be better described. Which enzyme was used? How was the amplified product detected? Although it is briefly mentioned in the results and discussion section, it should be clearly described in material and methods section.

Table 2. What does P (%) stand for on that table?

Reviewer 2 Report

Overall, I think the argument of this paper reflects what it is requested by the field and what it is needed to overcome the spreading of contagious agalactiae within a herd. This paper is well written, and all experimental works are well done. I have only minor comments on what could be adjusted and improved.

  • Line 37: "The optimal temperature-time combination of amplification was 64ºC in 60 minutes". It is not necessary to include this sentence in the abstract.
  • Line 61: Include a reference
  • Line 81: Shortly describe the difference between PCR amplification and LAMP amplification.
  • Line 90: Specify if the method tested and under evaluation is commercial or if it is an in-house developed or if it was previously developed by other groups. 
  • Line 95: Are there available details on how the wild strains have been identified?
  • Line 106: Use "spiked" instead of "contaminated"
  • Line 113-117: "As reference test, we have compared LAMP assay (ICGene® ) to a well-established real-time PCR method (Biorad) for the amplification of a DNA region of M. agalactiae p40 gene encoding an immunodominant adhesin that plays a crucial role in mycoplasma virulence. Genomic DNA was 116 extracted from 1 ml of contaminated milk, using the InstaGeneTM Matrix (Biorad) according to the 117 manufacturer’s instructions." This sentence could be also removed since these informations are then repeated. Some procedure details would need to be redistributed. The importance of p40 as a gene target could be also moved in the introduction. In the material and methods, I would write a paragraph with the title "Samples preparation' and here report details on how spiked samples have been prepared and how DNA has been extracted. 
  • Line 116: use "spiked milk" instead of "contaminated milk"
  • Line 135: Specify if specific software or programs have been used to conduct statistical analysis.
  • Line 140: "In fact, primers have been designed to detect with high
    specificity only the M. agalactiae DNA without cross-reaction to other pathogenic mycoplasma", this sentence seems not necessary here,  it could be moved somewhere else or it could be rephrased to fit better on the context of the paragraph.
  • Line 151: "detection time" could be misunderstood, use processing time
  • Line 149-153: here reported two topics, first sensitivity and then the processing time requested before results are finalized. These two topics are crucial for the comparison of the two assays here evaluated and for this reason, this paragraph should be reformulated to be more clear.
  • Line 173: The usage of the word "speed" seems to not be the best choice for the context. "turnaround time" is the technical term used most of the time. 

Reviewer 3 Report

You must include the STARD checklist and follow it through covering all elements o the list (target and source population, sample size, blindness etc).

Abstract

Here and elsewhere: all estimates must have confidence intervals

A better presentation of your main outcomes and conclusion is needed here

Introduction

Too many paragraphs

Materials and Methods

Description of your statistical methodology must be seriously improved. Please see relevant published work.

A description of the target and the source population is missing. Is the target and the source population the same or not?  Where can you extrapolate your results?

You could describe your statistical methodology in a more transparent wa.

Briefly give sample size calculations that are specific to this risk factor study.

Blindness: were the people performing the one test blinded to results of the other test?

Results

Please provide confidence intervals for all estimates.

Results should be (whenever feasible) in series of appearance that matches the appearance of the corresponding methods in the M&M section. 

Discussion

Discussion is short of references and the discussion of your findings in relation to similar work in the field.

Tables and figures

Tables and figures need improvement. Abbreviations defined for the first time should not be repeated. Tables and figures should stand alone but once something has been defined and abbreviated you need not repeat it afterwards.
